# Identification of NOL-Ring Composite Materials’ Damage Mechanism Based on the STOA-VMD Algorithm

**DOI:** 10.3390/polym15122647

**Published:** 2023-06-11

**Authors:** Peng Jiang, Hui Li, Xiaowei Yan, Luying Zhang, Wei Li

**Affiliations:** 1College of Mechanical Science and Engineering, Northeast Petroleum University, Daqing 163318, China; jpnepu2022@163.com (P.J.); treasurelh23@163.com (H.L.); liweinepu2021@163.com (W.L.); 2Shandong Gaint E-Tech Co., Ltd., Jinan 250000, China; yanxwmaster@163.com

**Keywords:** acoustic emission, glass fiber/epoxy NOL-rings, composite materials, matrix cracking, variational mode decomposition, pattern recognition

## Abstract

This research utilized the sooty tern optimization algorithm–variational mode decomposition (STOA-VMD) optimization algorithm to extract the acoustic emission (AE) signal associated with damage in fiber-reinforced composite materials. The effectiveness of this optimization algorithm was validated through a tensile experiment on glass fiber/epoxy NOL-ring specimens. To solve the problems of a high degree of aliasing, high randomness, and a poor robustness of AE data of NOL-ring tensile damage, the signal reconstruction method of optimized variational mode decomposition (VMD) was first used to reconstruct the damage signal and the parameters of VMD were optimized by the sooty tern optimization algorithm. The optimal decomposition mode number K and penalty coefficient α were introduced to improve the accuracy of adaptive decomposition. Second, a typical single damage signal feature was selected to construct the damage signal feature sample set and a recognition algorithm was used to extract the feature of the AE signal of the glass fiber/epoxy NOL-ring breaking experiment to evaluate the effectiveness of the damage mechanism recognition. The results showed that the recognition rates of the algorithm in matrix cracking, fiber fracture, and delamination damage were 94.59%, 94.26%, and 96.45%, respectively. The damage process of the NOL-ring was characterized and the findings indicated that it was highly efficient in the feature extraction and recognition of polymer composite damage signals.

## 1. Introduction

To meet the growing demand for material properties in structural applications, fiber-reinforced composites are widely used in automotive, aerospace, wind power, biomedical, and other fields because of their excellent properties such as a light texture, high strength, and good fatigue resistance [1,2]. However, different forms of damage occur whilst manufacturing and using composite materials, affecting the normal use of the structure. Therefore, the characterization, monitoring, and accurate identification of damage types are extremely important to understand how damage occurs [3,4,5,6,7].

Acoustic emission (AE) is related to the transient elastic wave generated by the redistribution of the stress field. When a structure changes, the energy is transmitted to its surface as waves. AE is a real-time, continuous, online monitoring, and nondestructive testing method. It is widely used in determining the damage to composite materials [8]. Composite materials demonstrate distinct characteristics at various stages of damage. AE signals can be used to characterize different failure mechanisms such as matrix cracking, interface failures, and fiber fractures [9,10,11,12,13] and each damage mechanism is associated with specific AE characteristic parameters (amplitude, count, and energy) [14,15,16]. The research on signal analysis and processing methods is one of the key issues in the identification and evaluation of acoustic emission sources. For instance, Jiang et al. [17] employed the extracted modal characteristics of damage. They established modal acoustic emission parameters to determine the damage mode of carbon fiber/epoxy resin composite pressure vessels. They also conducted a modal characterization of the acoustic emission signal associated with damage, facilitating the identification and characterization of the material damage mechanism. Additionally, Ghaib et al. [18] utilized acoustic emission technology to investigate the damage evolution of glass-fiber-reinforced polymer (GFRP) composite plates in bending tests. They calculated time and frequency domain parameters during the experiments and proposed an original method implemented in MATLAB software to determine the parameters characterizing each signal. Djabali et al. [19] utilized the acoustic emission technique and digital image correlation (DIC) to examine the mechanical and damage behavior of a thick laminated carbon/epoxy composite. They conducted static tests and three-point fatigue bending experiments while analyzing acoustic emission signal parameters such as amplitude, number of counts, duration, and cumulative AE energy. Joselin et al. [20] demonstrated in their study on glass fiber/epoxy resin composite material damage experiments that the NOL-ring experiment subjected the specimen to circumferential stress, which is similar to the stress induced by internal pressure between confined regions. Compared with a standard tensile experiment, the NOL-ring experiment was closer to the stress distribution characteristics of the pressure vessel and found serious and accidental pressure vessel damage more effectively. For instance, Plöckl et al. [21] employed unsupervised pattern recognition technology to establish the correlation between AE signals and mechanisms such as matrix cracking, interface failures, and fiber fractures. They monitored AE signals during the loading of NOL-ring specimens composed of carbon fiber and the thermoplastic polymer polyphenylene sulfide (PPS). Therefore, selecting NOL-ring specimens as the research object was effective in establishing the relationship between acoustic emission signals and the methods/mechanisms of damage identification.

Signal processing and the extraction of relevant signal features are crucial for the processing of materials and condition monitoring. Dragomiretskiy et al. [22] introduced the variational mode decomposition (VMD) algorithm, which effectively decomposes signal frequency components and exhibits excellent noise reduction capabilities. Unlike other classical adaptive mode decomposition methods such as empirical mode decomposition (EMD) and a high-voltage differential (HVD), VMD has minimal practical and theoretical limitations. For instance, Civera et al. [23] conducted a comprehensive comparison of complete ensemble empirical mode decomposition with adaptive noise (CEEMDAN), HVD, and VMD for the adaptive mode decomposition of vibration-based structural health-monitoring signals. Based on their evaluation, VMD was recommended as the most viable choice. VMD is highly effective in analyzing signal frequency distributions and extracting signal characteristics. For example, Yuan et al. [24] proposed a tool-wear monitoring method that combined VMD with ensemble learning. Bazi et al. [25] monitored tool wear using VMD in conjunction with the hybrid convolutional neural networks–bidirectional long short-term memory (CNN-BiLSTM) approach. Wan et al. [26] proposed a signal reconstruction method based on parameter adaptive VMD to accurately differentiate and identify various wear states of ceramic grinding wheels. To enhance the recognition performance of bearing fault signals, Li et al. [27] applied the genetic algorithm (GA) to optimize a combination of VMD parameters, resulting in the GA-VMD algorithm that improved the decomposition accuracy of VMD. This approach enabled the accurate recognition of different bearing fault signals under multifeature conditions. However, their study primarily focused on data-driven feature extraction methods and algorithms utilizing VMD and applied them to the fault damage identification of mechanical equipment such as tools or bearings. The relationship between signal characteristics and the damage process of composite materials established through VMD remains scarcely studied. Building upon an improved adaptive time–frequency analysis algorithm, Cao et al. [28] combined EMD, a correlation coefficient analysis, a fuzzy entropy algorithm, and a Hilbert transformation to enhance the depth evaluation of the phased-array ultrasonic nondestructive testing of composite structures. Consequently, further research is necessary to determine the relationship between signal characteristics and the corresponding damage categories in the process of polymer composite material damage through VMD.

It is usually necessary to combine VMD with other methods for the diagnosis of the state of damage. Compared with empirical mode decomposition [29], VMD transforms signal decomposition into a nonrecursive VMD mode, which has a solid theoretical foundation, and its variational decomposition effect is greatly affected by the number of modes K and the penalty coefficient α. Based on VMD, K modes are obtained. The correlation coefficient method, energy method, entropy method, distance method, and parameter optimization method can be utilized to reconstruct and denoise modes [30,31,32], leading to more precise and efficient feature analyses. Ram et al. [33] conducted a study where kurtosis of the envelope signal served as an indicator to optimize the number of modes and select the most suitable intrinsic mode function (IMF). In another investigation, Li et al. [34] used kurtosis as the index to optimize the mode number K and penalty factor α of VMD, and the optimal IMF was selected according to the resonance frequency. The kurtosis value of the envelope signal was used to determine the optimal mode number and penalty factor. In another study, Li et al. [35] used signal envelope entropy as the objective function. They obtained optimal parameters by using the optimization algorithm to optimize the number of decomposition modes and penalty coefficients to perform feature extraction and analyze bearing faults. Yao et al. [36] combined the sooty tern optimization algorithm (STOA), VMD, a support vector machine (SVM), and a backpropagation neural network (BPNN) to construct a financial data prediction fusion model based on decomposition–recombination, which effectively improved the prediction accuracy of financial time series.

Based on the above findings, in this study, we optimized the VMD parameters using the STOA, determined the optimal decomposition mode number K and penalty coefficient α, and performed an adaptive decomposition of acoustic emission signals. Based on the characteristics of a single damage signal, the corresponding relationship between the acoustic emission signals of different frequency components and damage categories was established and a damage mechanism recognition algorithm was constructed and verified. Finally, the identification algorithm was used to analyze a glass fiber/epoxy NOL-ring tensile experiment to evaluate the effectiveness of the algorithm and determine the mechanism of damage. This further improved the efficiency and accuracy of the signal feature extraction and recognition of polymer composites, realizing the state judgment and characterization of polymer composites.

## 2. Methods

### 2.1. Overview of VMD

Based on the Wiener filtering theory, VMD was formulated as a new time–frequency signal decomposition method in 2014 [22]. It has a stronger anti-noise ability than classical algorithms such as local mean decomposition and empirical mode decomposition, and can construct and solve variational problems. The principle and derivation process of VMD can be summarized as follows:The original signal is decomposed into K-independent modes *u_K_*(*t*); the Hilbert transform is then performed to obtain the unilateral spectrum.The decomposition sequence is a finite bandwidth modal component with a central frequency.The corresponding constrained variational model can be expressed as shown in Equation (1).
(1)$$\left\{\begin{array}{c} min\left\{\sum_{K}{\left\|\partial_{t}\left[\left(\delta \left(t\right)+\frac{j}{\pi t}\right)*u_{K}\left(t\right)\right]e^{-j\omega_{K}t}\right\|}_{2}^{2}\}\right.\\ s.t.\sum_{K}u_{K}=f \end{array}\right.$$

In the equation, $\left\{u_{K}\right\}$ indicate the K^th^ modal component, $\left\{\omega_{K}\right\}$ indicate the center frequency, respectively, and ∗ represents the convolution operator.

4.The constrained variational problems are transformed into unconstrained variational problems. The augmented function can be introduced, as depicted in Equation (2).


(2)
$$L\left(\left\{\underset{K}{u}\right\},\left\{\omega_{K}\right\},\lambda \right)=\alpha \sum_{K}{\left\|\partial_{t}\left[\left(\delta \left(t\right)+\frac{j}{\pi t}\right)*u_{K}\left(t\right)\right]e^{-j\omega_{K}t}\right\|}_{2}^{2}+{\left\|f\left(t\right)-\sum_{K}u_{K}\left(t\right)\right\|}_{2}^{2}+\langle \lambda \left(t\right),f\left(t\right)-\sum_{K}u_{K}\left(t\right)\rangle$$


In the equations, α presents the quadratic penalty factor and $\lambda \left(t\right)$ is the Lagrange multiplication operator.

5.The alternating direction multiplier method (ADMM) is used to find the ‘saddle point’ of the augmented function. The models of *u_K_* and *ω_K_* after alternating the optimization iterations are as follows:


(3)
$${\hat{u}}_{K}^{n+1}\left(\omega \right)=\frac{\hat{f}\left(\omega \right)-\sum_{i=1,i\neq K}^{K}\hat{u}\left(\omega \right)+\frac{\hat{\lambda }\left(\omega \right)}{2}}{1+2\alpha {\left(\omega -\omega_{K}\right)}^{2}}$$



(4)
$$\omega_{K}^{n+1}=\frac{\int_{0}^{\infty }\omega {\left|{\hat{u}}_{K}(\omega )\right|}^{2}d\omega }{\int_{0}^{\infty }{\left|{\hat{u}}_{K}(\omega )\right|}^{2}d\omega }$$


In the equations, $\omega_{K}^{n+1}$ represents the center of the power spectrum of the current modal function, ${\hat{u}}_{K}^{n+1}\left(\omega \right)$ is equivalent to the Wiener filtering of the current residual $\hat{f}\left(\omega \right)-\sum_{i\neq K}\hat{u_{i}}(\omega )$, and Fourier transform is performed on $\left\{{\hat{u}}_{K}\left(\omega \right)\right\}$.

### 2.2. The VMD Parameters K and α Can Be Optimized Based on the STOA

In 2019, Dhiman et al. [37] and other research groups proposed an optimization algorithm, STOA, for solving industrial engineering problems. This algorithm can be used to conduct a global search and it has high data-processing accuracy. Based on the envelope spectrum, we established the minimum envelope entropy as a moderate function to improve the STOA and optimize the VMD parameters K and α. The entropy value can be used to measure the uncertainty and complexity of a damage signal. As the entropy value becomes larger, the damage signal becomes more complex.

The envelope entropy of component ${IMF}_{i}(K)$ after VMD can be expressed as:(5)$$\left\{\begin{array}{c} E_{i}=-\sum_{K=1}^{n}P_{i,K}lgP_{i,K}\\ P_{i,K}=a_{i}(K)/\sum_{K=1}^{N}a_{i}(K) \end{array}\right.$$

In the equation, i is the serial number of IMF after the decomposition of the original signal $y\left(K\right)(i=1,2,3\cdot \cdot \cdot \cdot$); $a_{i}(K)$ is normalized to $P_{i,K}$ and $a_{i}(K)$ is the envelope signal of the signal ${IMF}_{i}(K)$ after the Hilbert transform.

First, the parameters of the STOA were initialized. The number of the initial population was 30 and the number of iterations was 50.

The position of the tern was reinitialized as follows: set $K\in \left[2,10\right],\alpha \in \left[1000,4000\right]$. Both were random values.

The main steps were as follows [37]:


The individual fitness value and population average fitness were calculated.Migration behavior was observed or a global search was performed.a. Crash avoidance:




(6)
$$c_{st}=S_{A}\times P_{st}(Z)$$



In the equation, $P_{st}$ denotes the current position of the tern, $c_{st}$ denotes the new position of the individual, $S_{A}$ denotes a variable parameter for collision avoidance, and $S_{A}$ could be updated as follows:$$S_{A}=C_{f}-\left(Z\times \left(\frac{C_{f}}{{Max}_{iterations}}\right)\right),Z=0,1,\cdot \cdot \cdot ,{Max}_{iterations}$$

Here, $C_{f}=2$; Z was the current number of iterations.

b. Determination of the relative spacing:



(7)
$$m_{st}=C_{B}\times (P_{bst}\left(Z\right)-P_{st}\left(Z\right))$$



In the equation, $m_{st}$ denotes the relative distance between the current individual and the optimal individual, $P_{bst}$ denotes the position of the current optimal individual, and $C_{B}$ is a more comprehensive exploration of the random position $C_{B}=0.5\times R_{and}$, where $R_{and}\in$ [0, 1].

c. If close to the optimal individual:



(8)
$$d_{st}=c_{st}+m_{st}$$




3.Attack behavior was observed or a local search performed. The mathematical models for the attack behavior were:




(9)
$$\{x^{\prime }==R_{adius}\times sin(i)y^{\prime }=R_{adius}\times cos(i)z^{\prime }=R_{adius}\times i$$


(10)
$$R_{adius}=u\times e^{Kv}$$



Here, $i\in \left[0,2\pi \right];R_{adius}$ represents the radius of each spiral and u=v=1 defines the spiral shape.

4.The final position update of the tern was acquired, as follows:


(11)
$$P_{st}(Z)=(d_{st}\times \left(x^{\prime }+y^{\prime }+z^{\prime }\right)\times P_{bst}(Z)$$


5.The fitness value was calculated and the global optimal value was retained.

The flowchart of the algorithm optimization parameters is shown in Figure 1.

## 3. Experiments and Results

### 3.1. Experimental Materials and Methods

#### 3.1.1. Composite Laminates

The matrix specimens in this study were dumbbell-shaped epoxy resin castings (250 mm × 25 mm × 25 mm; length × width × height, respectively). The performance parameter test of a glass-fiber bundle (SC-1200) in the performance test of the glass fiber multifilament dipping method was described in GB/T 7690.3–2013 [38] “Reinforcements-Test method for yarns-Part 3: Determination of breaking force and breaking elongation of glass fiber”. The GB/T 3362–2017 [39] “Test method for tensile properties of carbon fiber multifilament” standard was followed to establish the performance parameters of the material, as presented in Table 1. The prepreg consisted of a glass-fiber fabric and epoxy resin, with an approximate volume fraction of 80% TDE-85# epoxy resin. The curing process involved autoclaving the prepreg at 120 °C for 3 h, followed by heating it to 160 °C for 3 h and subsequently raising the temperature to 180 °C for 4 h. A pressure of 0.2 MPa was applied to the prepreg to ensure the removal of air and volatiles without excessive resin squeezing. After cooling to room temperature, the glass/epoxy composite laminates were cut into the desired dimensions. A 1200 Tex high-strength S-glass-fiber prefabricated laminated composite plate (175 mm × 25 mm × 4 mm; length × width × height, respectively) was used to simulate the delamination damage experiment according to the ASTM D5528 standard [40].

#### 3.1.2. NOL-Ring

In the tensile acoustic emission monitoring experiment of the NOL-ring, an SC-1200 high-strength glass-fiber-reinforced material was used to prepare the glass fiber 1200Tex NOL-ring specimen, as described in GB/T 1458–2008 “Test method for mechanical properties of the ring of filament-winding reinforced plastics”. The prepreg consisted of a glass-fiber fabric and epoxy resin, with an epoxy resin TDE-85# volume fraction of approximately 80% and a fiber volume content of at least 58%. The NOL-ring specimen was 150 mm × 6 mm × 1.5 mm (diameter × width × thickness) and the winding tension was 20 N. While conducting the static tensile test of the NOL-ring, there was mutual friction and vibration interference between the NOL-ring and the fixture. Therefore, a NOL-ring grading loading experiment was conducted to simulate the cyclic process of loading–holding–loading, in which the holding stage was maintained for 4 min. The graded loading experimental scheme is shown in Figure 2.

#### 3.1.3. Experimental Equipment Settings

Based on the assumption that different damage modes produce acoustic emission signals with specific characteristics, the experiment was divided into single damage acoustic emission monitoring and NOL-ring tensile acoustic emission monitoring to identify different damage modes. The experimental equipment included a PCI-E acoustic emission detector (MICRO-Ⅱ EXPRESS, Physical Acoustics Corporation, 195 Clarksvikke Road Princeton Junction, NJ 08550), an SEMTester (MTI Instruments, 325 Washington Ave Ext Albany, NY 12205) in situ tensile testing machine, and a Shimadzu AG-X electronic universal testing machine (AG-XD 20KN, SHIMADZU(CHINA) Co., Ltd., Shanghai, China). The AE monitoring system utilized in all experiments consisted of the AE software AE-Win for Express-8 (V5.92) and the acquisition module for recording AE signals, which was provided by the Physical Acoustics Corporation (PAC), Princeton Jct, New Jersey, USA. The equipment used in the experiment is shown in Figure 3. The sampling rate of the PCI-E acoustic emission detector was 1 Million samples per second (MSPS). The peak definition time (PDT), hit definition time (HDT), and hit lockout time (HLT) were 100, 200, and 400, respectively. The parameters are shown in Table 2.

### 3.2. Results

#### 3.2.1. Analysis of the Acoustic Emission Parameter History

The acoustic emission process diagram of single damage was analyzed and the acoustic emission parameters of each damage form are shown in Figure 4, Figure 5 and Figure 6. The amplitude of the epoxy resin matrix tensile test was mainly distributed around 40–60 dB and the step phenomenon occurred in the energy accumulation curve, with the largest impact at a later stage of the tensile test. The tensile amplitude of the fiber bundle was mainly distributed around 75–95 dB. At the stage around 18–22 s, a large number of high-amplitude and high-energy AE signals appeared. The layered signal was mainly concentrated around 45–70 dB and the ringing count was concentrated around 40–75 dB. The cumulative energy curve linearly increased, indicating that the damage mechanism was single.

In this study, the amplitude, duration, cumulative count, and energy were selected as acoustic emission parameters to analyze the distribution of the glass fiber/epoxy NOL-ring cyclic loading process. Different degrees of visible damage occurred at the initial load mutation position in the third, sixth, and eighth holding stages (Figure 7). In the initial stage of stretching, the energy and duration were low and the amplitude was mainly below 65 dB. The cumulative energy curve showed that the curve increased stepwise in the later stage of the experiment and the amplitude was considerably higher than 85 dB. From the perspective of amplitude, the high count and high energy met the signal characteristics of fiber breakage.

#### 3.2.2. Optimization Analysis of the Signal Parameters of the NOL-Ring Step Loading Tensile Test

Optimizing the VMD parameters K and α

The two matrix cracking signals that we examined were randomly selected as samples. The STOA-VMD parameters were then optimized and the average values of multiple optimization parameters were calculated. The VMD signal parameters were determined by combining the center frequency of VMD and the data in Table 3. From the optimized results of the two samples, we concluded that the final selection should be K = 9 and α = 2950.

2.Comparison diagram of time–frequency analysis of matrix cracking signal

The acoustic emission signal resulting from matrix cracking in the damage exhibited medium-to-low intensity, while the delamination signal in the composite damage was challenging to distinguish and fell within the medium intensity range. On the other hand, the fiber fracture signal was classified as medium-to-high intensity. A tensile test conducted on a single material showed that the matrix cracking signal was distinct and exhibited minimal noise such as mechanical friction. Hence, the matrix cracking signal was chosen for the analysis. Based on this selection, the signal of sample 1 underwent VMD. The time domain and frequency domain plots of the decomposed signal are presented in Figure 8. Observing Figure 8b,d, it became apparent that the original signal after VMD produced nine signal components, and they were represented by different colors. Each component was primarily confined to a different frequency range, indicating a highly effective decomposition outcome.

3.Comparative analysis of the single damage reconstruction signal

From the frequency domain diagram of single damage, including matrix cracking, interface delamination, the original signal in the fiber fracture, and the reconstructed signal after the Fourier transform, we found that the reconstructed signal effectively suppressed the components, with a low correlation between the high-frequency and low-frequency domains, and distinguished different damages in the frequency domain. The frequency domain diagrams of the original and reconstructed signals of every single damage are shown in Figure 9.

According to the above findings, we established high and low band-pass filters based on reconstructed frequency domain features. The low-pass filter (LF) and high-pass filter (HF) were set to distinguish delamination damage. According to the distribution of matrix cracking and fiber fractures in different frequency bands, the high-pass filter was divided into HF1 and HF2. The specific process of judging the damage category is shown in Figure 10.

#### 3.2.3. Identification and Verification of the Damage Optimization Algorithm

Identification and verification of the single damage optimization algorithm

Using the recognition algorithm proposed in this study, a single damage was identified to verify the accuracy and effectiveness of the algorithm. As shown in Figure 11, the cumulative impact curve of the single damage identification was similar to the cumulative energy curve of a single experiment, indicating that the algorithm could effectively identify the single damage mechanism. The recognition results of the algorithm are shown in Table 4. The accuracy rates of matrix cracking, fiber fractures, and delamination damage were 94.59%, 94.26%, and 96.45%, respectively.

2.Identification and verification of the NOL-ring damage optimization algorithm

The cumulative curve of AE parameters in the glass fiber/epoxy NOL-ring tensile test was similar to the cumulative curve of damage identification (Figure 7 and Figure 12) and a step phenomenon was found in the ninth holding stage, which corresponded with the severe damage in the later stage of the NOL-ring tensile test. In the tensile part of the tenth stage, the curve showed a step growth phenomenon that corresponded with the serious increase in damage in the later stage of the experiment. The number of typical damage events identified at different stages is shown in Table 5 and Figure 13. Throughout the tensile test stage of the NOL-ring, the proportion of the matrix cracking impact number was large. The proportion of the matrix cracking impact number was about 53% during tensile stages 5–8; it first decreased and then increased. On the contrary, the number of fiber breakages first increased and then decreased. As the load increased, after a certain local failure, the load was redistributed and transmitted to the adjacent fibers through the matrix, causing fiber breakage. The number of interface layers significantly decreased in tensile stages 3–6; it first decreased and then increased.

## 4. Conclusions

In this study, a damage classification and recognition mechanism of the NOL-ring was proposed and the optimization algorithm of STOA-VMD was used to extract damage acoustic emission signals of polymer composites and recognize its damage state. First, the optimal decomposition mode number K and penalty coefficient α were determined based on the STOA to realize the adaptive decomposition of acoustic emission signals; then, a feature sample set was constructed according to the characteristics of the damage signal of a single material tensile test. Next, the accuracy of the optimization algorithm was evaluated. Finally, acoustic emission signal feature extraction was applied to the glass fiber/epoxy NOL-ring breakage experiment for damage classification and recognition and the damage state identification and characterization of the polymer composites were carried out. The main conclusions were as follows:The STOA was used to optimize the VMD parameters, determine the optimal decomposition mode number K and penalty coefficient α, and perform the adaptive decomposition of acoustic emission signals.Based on the characteristics of a single damage signal, the damage mechanism recognition algorithm was constructed and the recognition effect was verified. The recognition rates of matrix cracking, fiber fractures, and delamination damage were 94.59%, 94.26%, and 96.45%, respectively.By analyzing the AE signal of the glass fiber/epoxy NOL-ring experiment, the algorithm was used to determine the damage mechanism of the NOL-ring experiment, which confirmed that the algorithm could effectively perform damage identification of the glass fiber/epoxy NOL-ring damage complete structure.

In summary, the optimization algorithm based on STOA-VMD offered a more precise and efficient method to identify and characterize damage in polymer composites, thus serving as a valuable technical reference for researchers involved in the monitoring and characterization of mechanical structures or advanced polymeric materials. In future studies, the authors intend to explore improved algorithm optimization alternatives and conduct further comparative analyses. Additionally, efforts will be made to establish a scientific evaluation system that enables the condition monitoring, identification, and evaluation of polymer materials. Ultimately, this will facilitate a comprehensive and effective objective assessment of the state of polymer composite materials.

## Figures and Tables

**Figure 1 polymers-15-02647-f001:**
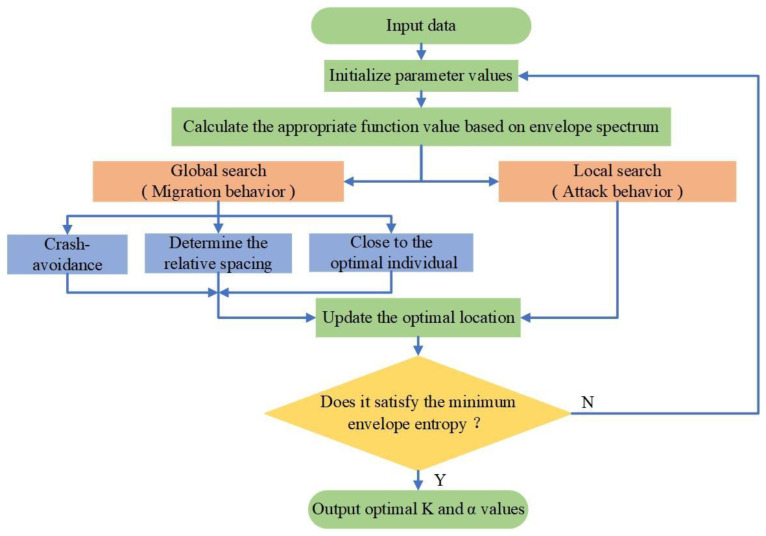
The flowchart of optimized VMD parameters based on the STOA.

**Figure 2 polymers-15-02647-f002:**
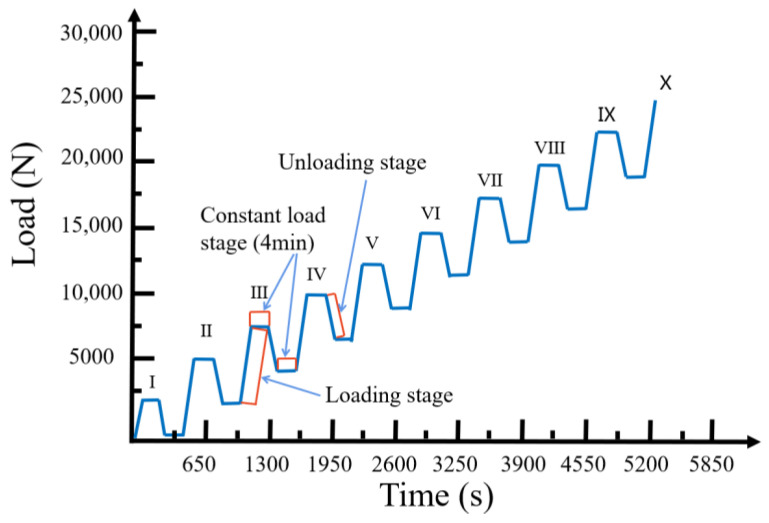
The NOL-ring graded loading experimental scheme.

**Figure 3 polymers-15-02647-f003:**
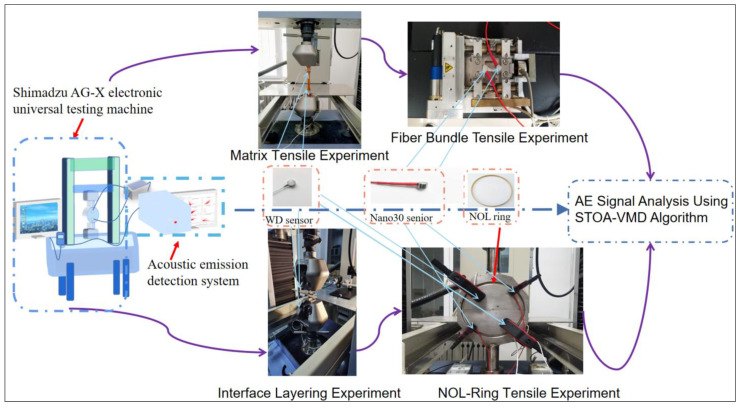
The equipment used in the experiment.

**Figure 4 polymers-15-02647-f004:**
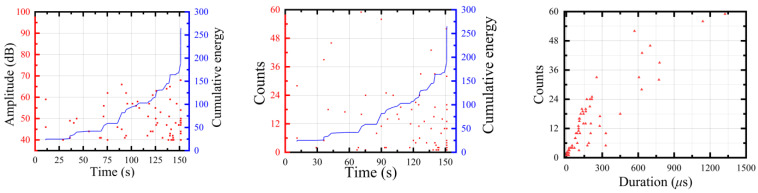
The matrix tensile acoustic emission parameter history.

**Figure 5 polymers-15-02647-f005:**
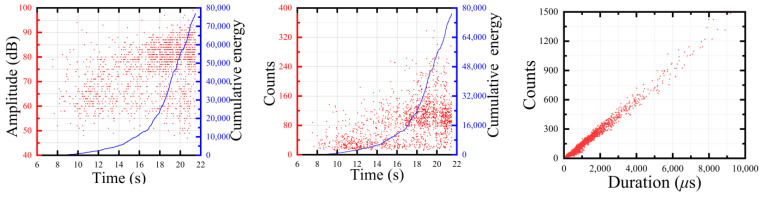
The tensile acoustic emission parameter history of glass-fiber bundles.

**Figure 6 polymers-15-02647-f006:**
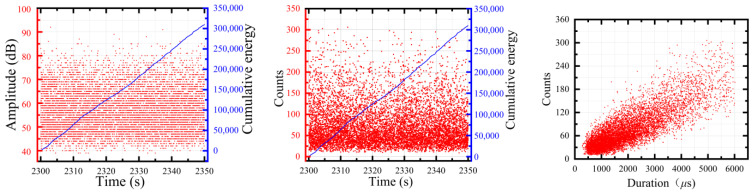
The interface layer acoustic emission parameter history.

**Figure 7 polymers-15-02647-f007:**
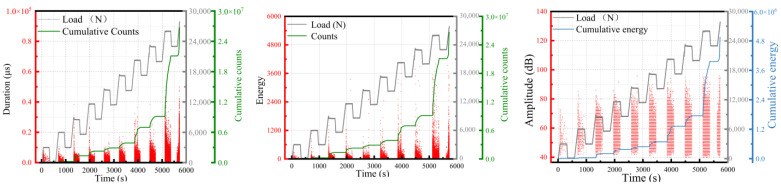
The analysis of the acoustic emission parameters of the NOL-ring step loading tensile test.

**Figure 8 polymers-15-02647-f008:**
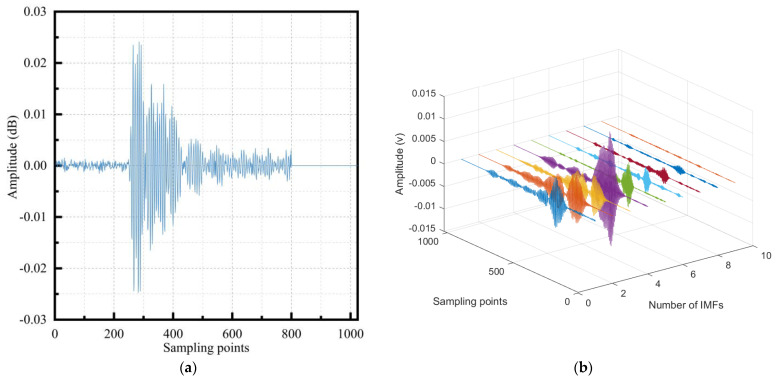
Time domain and frequency domain analysis of matrix cracking: (**a**) time domain diagram of the original signal; (**b**) time domain diagram of VMD; (**c**) frequency domain diagram of the original signal; (**d**) frequency domain diagram of VMD.

**Figure 9 polymers-15-02647-f009:**
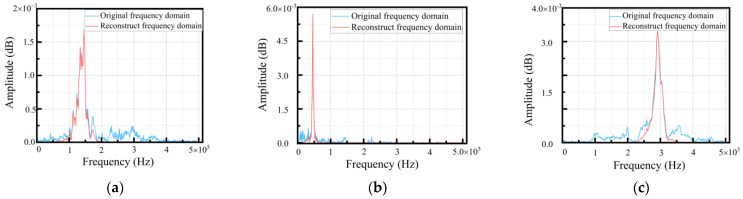
Comparison of original and reconstructed frequency domain of single damage: (**a**) matrix cracking signal; (**b**) interface delamination signal; (**c**) fiber fracture signal.

**Figure 10 polymers-15-02647-f010:**
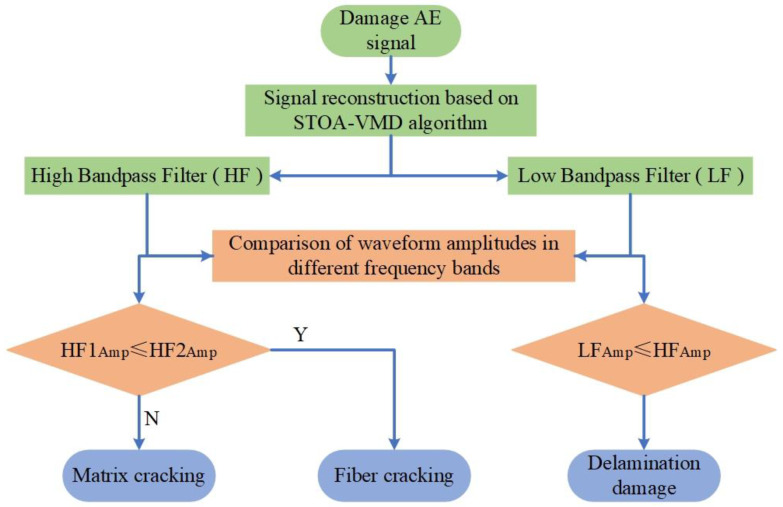
A flowchart of damage classification and recognition algorithm.

**Figure 11 polymers-15-02647-f011:**
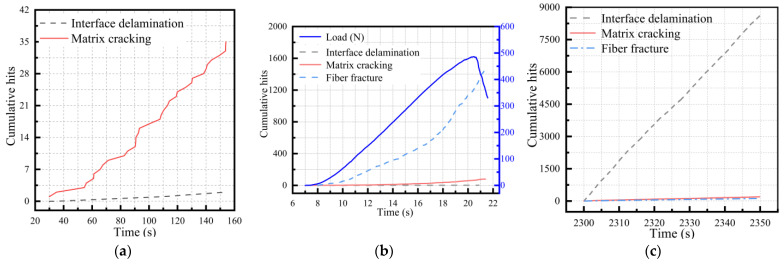
The identification diagram of the single damage algorithm: (**a**) matrix cracking; (**b**) fiber fracture; (**c**) interface delamination.

**Figure 12 polymers-15-02647-f012:**
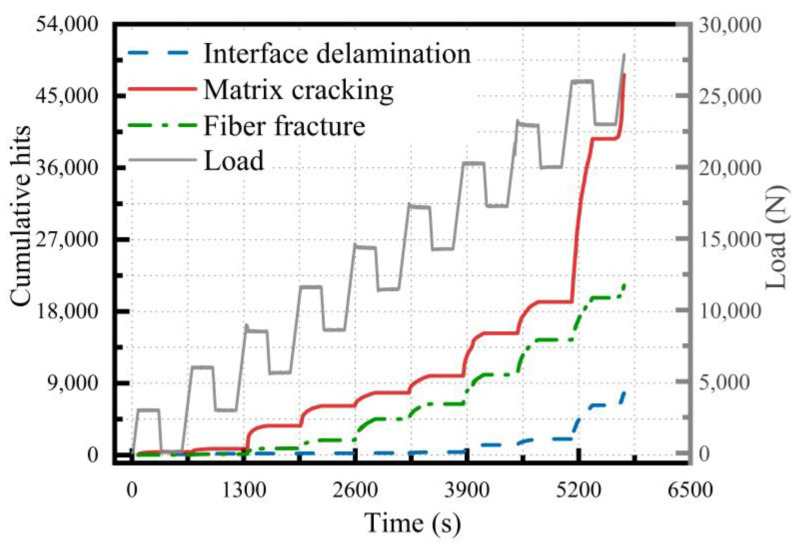
Damage identification diagram of the NOL-ring.

**Figure 13 polymers-15-02647-f013:**
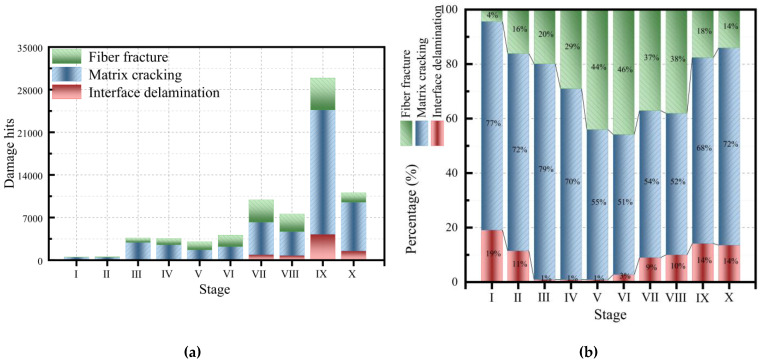
Damage accumulation diagrams of the NOL-ring at different stretching and holding stages: (**a**) damage accumulation histogram; (**b**) damage cumulative percentage figure.

**Table 1 polymers-15-02647-t001:** Performance parameters of materials.

Type Parameter	Tensile Strength/GPa	Tensile Modulus/GPa	Elongation (%)	Density (kg/m^3^)	Poisson Ratio
Matrix	0.12	3	0.2	980	0.38
Glass-fiber bundles	0.28	90	3.5	1800	0.3

**Table 2 polymers-15-02647-t002:** The list of parameters for the acoustic emission system.

Item	Sensor Type	Number Field of Channel (s)	Threshold/dB	Pre-Trigger Time/μs	Signal Length
Matrix tensile	WD	2	35	256	1024
Glass-fiber bundle tensile	Nona30	2	35	256	1024
Interface layering	WD	2	35	256	1024
NOL-ring tensile	WD	2	35	256	1024
	Nona30	2			

**Table 3 polymers-15-02647-t003:** The STOA parameters and the EMD mode number.

Type	EMD Decomposition Mode Number	Times	1	2	3	4	Average Value
Sample 1	11	K	9	10	8	9	9
		α	2856.670	2574.752	3551.249	2786.009	2950.084
Sample 2	10	K	8	8	9	8	8.25
		α	3446.525	1243.647	3110.164	4000	2942.170

**Table 4 polymers-15-02647-t004:** The results of the single damage identification analysis.

Type	Matrix Tensile Test	Fiber Bundle Tensile Test	Interface Layering Test
Cumulative hits	37	1568	8964
Matrix cracking/pc	35	79	8628
Fiber fracture/pc	0	1478	197
Interface delamination/pc	2	2	121
Accurate recognition rate/%	94.59	94.26	96.45

**Table 5 polymers-15-02647-t005:** The damage identification of the NOL-ring.

Type	Stage 1	Stage 2	Stage 3	Stage 4	Stage 5	Stage 6	Stage 7	Stage 8	Stage 9	Stage 10
Damage hits	503	558	3627	3549	3013	4109	9901	7568	29,932	11,058
Interface delamination	96	64	34	32	26	113	890	758	4231	1499
Percentage/%	19.085	11.470	0.937	0.902	0.863	2.750	8.989	10.016	14.135	13.556
Matrix cracking	385	404	2871	2488	1660	2113	5342	3929	20,439	8005
Percentage/%	76.541	72.401	79.156	70.104	55.095	51.424	53.954	51.916	68.285	72.391
Fiber fracture	22	90	722	1029	1327	1883	3669	2881	5262	1554
Percentage/%	4.374	16.129	19.906	28.994	44.042	45.826	37.057	38.068	17.580	14.053

## Data Availability

Data can be provided upon request from the correspondence author.

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
