# Peer review of "Identification of NOL-Ring Composite Materials’ Damage Mechanism Based on the STOA-VMD Algorithm"

_polymers, 2023, doi:10.3390/polym15122647_

Round 1

Reviewer 1 Report

Authors need to address the following queries to improve the quality of the article.

1.       The article must undergo language proofreading, as I found grammatical errors.

2.       In the abstract, please write the full form of any abbreviation as it was mentioned for the first time.

3.       “showed that in the glass/epoxy resin” add glass fiber/epoxy resin.

4.       What is the type of epoxy resin used in the work? Epoxy has Tensile strength of 0.12 GPa, it means 120 MPa. Check the correctness for the same.

5.       Quality of all figures needs to be improved.

6.       Elaborate on the sub-figures of Figure 8.

7.       In the results, the author(s) is expected to try as much as possible to compare their results obtained with the existing or similar studies.

8.       Please add references for your discussion to support it.

9.       Kindly reconcile the conclusion with the study objectives.

10.   What are the practical implications of this study and the future directions? kindly state?

The article must undergo language proofreading, as I found grammatical errors.

Author Response

Dear reviewer,

Thank you very much for your comments and professional advice. These opinions help to improve academic rigor of our article. Based on your suggestion and request, we have made corrected modifications on the revised manuscript. Meanwhile,the manuscript had be reviewed and edited by a language service company. We hope that our work can be improved again. Furthermore, we would like to show the details as follows:

Comment 1:The article must undergo language proofreading, as I found grammatical errors.

Response: We are very sorry for the mistakes in this manuscript and inconvenience they caused in your reading. We have found a professional polishing agency to polish and modify the grammar of the manuscript. We hope it can meet the journal's standard. Thanks so much for your useful comments.

Comment 2: In the abstract, please write the full form of any abbreviation as it was mentioned for the first time.

Response: Thanks for your careful checks. We are sorry that our carelessness has brought you a bad reading experience. Based on your comments, we have written down the full form of abbreviations. Please see page 1 of the revised manuscript, lines 14.

Comment 3:“showed that in the glass/epoxy resin” add glass fiber/epoxy resin.

Response: We sincerely thank the reviewer for careful reading. As suggested by the reviewer, we have corrected the “glass/epoxy resin” into “glass fiber/epoxy resin” in whole paper.

Comment 4:What is the type of epoxy resin used in the work? Epoxy has Tensile strength of 0.12 GPa, it means 120 MPa. Check the correctness for the same.

Response: Thank you for pointing out this problem in manuscript.The material of epoxy resin is TDE-85 # and then we annotated it in the revised manuscript.Therefore, we have listed a table of mechanical parameters of epoxy resin matrix and glass fiber bundle required in single material tensile test as shown in Table 1. Please see page 6,lines 215 and lines 223.

Comment 5:Quality of all figures needs to be improved.

Response: Thanks for your suggestion.We have further improved the quality of graphics, including lines, font size, clarity, etc. We sincerely hope to meet the graphic standards of journals.

Comment 6:Elaborate on the sub-figures of Figure 8.

Response: Thank you for the above suggestion.

Comment 7:In the results, the author(s) is expected to try as much as possible to compare their results obtained with the existing or similar studies.

Response: Thank you for pointing this out in our manuscript. Here, the comparison of your proposed results with the existing or similar research results will indeed make our manuscript more perfect, but our research mainly uses the novel algorithm based on STOA-VMD proposed by us to realize the damage identification and characterization of polymer composites. Therefore, this paper compared the algorithm only by combining the results of single tensile test ( matrix tensile, fiber fracture, interface delamination ) and polymer NOL-ring tensile test. And because we found that there are few studies on the application of this method to polymer composites, mainly mechanical bearing or tool damage, we are sorry that we have not made a similar comparison with the existing research related to polymer composites. We sincerely hope to get your understanding and acceptance.

Comment 8:Please add references for your discussion to support it.

Response: Thanks for your suggestion. We 're sorry that we didn 't quite understand what you pointed out here to add reference to our discussion, but in the summary we added references to the background of the application of acoustic emission on-line monitoring technology to polymer composites. We hope to have the opportunity to get your further comment guidance to make our manuscript more perfect. see page 2, lines 51-60.

Comment 9:Kindly reconcile the conclusion with the study objectives.

Response: Thanks for your suggestion. We have rewritten some of the conclusions to make them more consistent with the research objectives. Please see page 15, lines 363-372.

Comment 10:What are the practical implications of this study and the future directions? kindly state?

Response: Thanks for your suggestion.Our research mainly provides a more accurate and efficient damage identification method for damage identification and state characterization of polymer composites. At the same time, it can provide a technical reference for scholars who are interested in the condition monitoring and characterization of mechanical structures or other advanced polymer materials. In the future, the author will further develop better algorithm optimization alternatives and conduct similar comparative analysis. Meanwhile, we are committed to establishing a scientific evaluation system to realize the condition monitoring, identification and evaluation of polymer materials, and to achieve a comprehensive, effective and objective state evaluation of polymer composites. Please see page 15, lines 385-394.

We would like to thank the reviewer again for taking the time to review our manuscriptpatiently and carefully.

Yours Sincerely,

Luying Zhang.

E-mail:zlyyy2023@163.com

Reviewer 2 Report

The manuscript "Identification of NOL-ring Composite Materials Damage Mechanism Based on the STOA-VMD Algorithm" is well written and the damage identification of the above algorithm reached high values. There only minor parts to address

1. It would be beneficial show in the introduction where such damage control find its application. Please add that

2. The algorithm the author used under what exception can be those applied and if having other materials is it suitable for such as well or only on the mention material?

3. Figure 7. The authors used maximum 85 dB. What happen above such? 

4. The authors used composite materials. Can such damage control as well be suitable for polymers in pure form? It would be beneficial adding some Table regarding to literature of damage control comparing their values to other made before.

The language is fine just minor spell checking

Author Response

Dear reviewer,

Thank you very much for your reading our paper carefully and giving the above positive comments. Based on your suggestions and requirements, we have added and modified the relevant contents of the manuscript. We hope that our work can be improved again. Furthermore, we would like to show the details as follows:

Comment 1:It would be beneficial show in the introduction where such damage control find its application. Please add that.

Response: We sincerely thank the reviewer for this suggestion.The damage identification and characterization method of polymer materials mentioned in this paper is mainly applied to mechanical bearing fault detection, tool wear and damage behavior of polymer composites.We have added some statements and related literature here, please see page 2-3, lines 78-104.

Comment 2:The algorithm the author used under what exception can be those applied and if having other materials is it suitable for such as well or only on the mention material?

Response: We thank the reviewer for pointing out this issue.Because our paper mainly optimizes the damage state and characterization of polymer composites under the background of VMD application based on mechanical bearing and tool wear acoustic emission signal damage state judgment. Then our algorithm is applied to the matrix tensile, glass fiber bundle tensile and interface delamination experiments in a single experiment, and also applied to glass fiber / epoxy resin polymer composites. It provides a more efficient and accurate method reference for the state and characterization of polymer composites, and has high guiding significance. At present, the VMD-based method is suitable for mechanical structures and polymer composites. As an improved VMD optimization algorithm, this paper is also applicable to the related research in the above fields.

Comment 3:Figure 7. The authors used maximum 85 dB. What happen above such?

Response: Thank you for pointing out this problem in manuscript.Thank you for pointing this out in your manuscript. When the amplitude reaches 85 dB, fiber breakage or large area interface delamination occurs, and fiber breakage dominates.

Comment 4:in pure form? It would be beneficial adding some Table regarding to literature of damage control comparing their values to other made before.

Response: Thanks for the reviewer’s comments sincerely. This damage identification method is suitable for pure polymers. This method was applied to the tensile test of epoxy resin matrix in our single material tensile test and the effect was good. However, because this optimization method based on STOA-VMD should be our first application to polymer composites, and there are few literatures on VMD algorithm for damage identification and characterization of polymer composites, there are no tables listed for comparison, hoping to get your understanding and acceptance. We sincerely hope to get your understanding and acceptance.

We would like to thank the reviewer again for taking the time to review our manuscript patiently and carefully.

Yours Sincerely,

Luying Zhang.

E-mail:zlyyy2023@163.com

Reviewer 3 Report

This study introduces the Sooty Tern Optimization Algorithm-Variational Mode Decomposition (STOA-VMD) optimization algorithm for extracting damage-induced acoustic emission (AE) signals in fiber-reinforced composite materials. The algorithm is validated using a glass/epoxy NOL-ring tensile experiment.

The proposal is overall interesting but several aspects should be all addressed before granting final acceptance. Specifically:

Major remarks:

1.     In the Reviewer’s opinion, the choice of VMD is indeed a very well-made decision. However, it should be better motivated. For instance, in https://doi.org/10.3390/s21051825, it was proved that VMD outperforms other common signal decomposition alternatives such as CEEMDAN and HVD for SHM-related tasks. This should be mentioned in the text.

2.      Related to the previous comment, as with many other data-adaptive decomposition algorithms, VMD is known to perform efficiently on structures and mechanisms with well-separated, clearly visible modes, while struggling with mixed or closely-spaced modes as well as weakly excited ones. It is not clear if the proposed benchmark problem falls into any of these compelling categories.

3.      Equation 2, the meaning of the operator < > should be explicitly stated.

4.      Optimising the VMD settings (K and alfa) is still a very relevant issue. The Authors propose here the use of the Sooty Tern Optimization Algorithm (STOA) for such purposes. However, this return two (slightly) different settings for sample 1 and 2. This Reviewer's issue is what would happen for a larger number of samples, i.e. if the averages would remain mostly similar or diverge.

Just as a note, the final result, alfa = 2000, is in line with what one would generally expect, i.e. something around alfa = 2950 for moderate bandwidth constraining.

5.      The paper should provide more details about the AE monitoring system and, in general, more context for Acoustic Emission-based Structural Health Monitoring, with applications to fiber-reinforced polymers (such as in https://doi.org/10.1016/j.compstruct.2020.113105) but also other materials.

Minor and editorial issues:

1.  Figure 3 seems to be ‘stretched’, plus the font size changes for different writings and some of them are too small to be easily readable.

2.    In general, for several figures (e.g. Figures 7 and 11) the font size of the x- and y-axis labels and ticks and/or the legend is too small.

3.   Table 5: is it really needed to report the percentage values up to the third decimal digit? That means an accuracy of 10 parts per million, which seems to be a bit excessive.

4.     It is not clear why the parameter K (uppercase) is sometimes written as k (lowercase).

According to what said above, the Reviewer’s opinion is that the manuscript can be accepted for publication after the described major revisions.

The English of the paper is overall good but grammar checking for potential typos is still recommended.

Author Response

Dear reviewer,

Thank you very much for your reading our paper carefully and giving the above positive comments.Based on your suggestions and requirements, we have added and modified the relevant contents of the manuscript. We hope that our work can be improved again. Furthermore, we would like to show the details as follows:

  1. Major remarks:

Comment 1:In the Reviewer’s opinion, the choice of VMD is indeed a very well-made decision. However, it should be better motivated. For instance, in https://doi.org/10.3390/s21051825, it was proved that VMD outperforms other common signal decomposition alternatives such as CEEMDAN and HVD for SHM-related tasks. This should be mentioned in the text.

Response:We are appreciative of the reviewer’s suggestion.According to the reviewer 's suggestion, we refer to the literature you mentioned and read it. A part of the content is supplemented later, and the EMD, HVD and VMD selected in this paper are analyzed and explained. Please see page 2, lines 79-86.

Comment 2:Related to the previous comment, as with many other data-adaptive decomposition algorithms, VMD is known to perform efficiently on structures and mechanisms with well-separated, clearly visible modes, while struggling with mixed or closely-spaced modes as well as weakly excited ones. It is not clear if the proposed benchmark problem falls into any of these compelling categories.

Response:Thanks for the reviewer’s comments sincerely. Our paper mainly optimizes the optimal decomposition mode number K and penalty coefficient alfa in traditional VMD based on STOA algorithm. The optimized algorithm effectively decomposes the appropriate modal frequency band range of each state acoustic emission signal, and then decomposes the acoustic emission signals of different frequency components. A data relationship was established between the damage category and the optimization algorithm. Finally, the constructed damage identification algorithm was applied to the tensile test of glass fiber / epoxy resin NOL-ring to verify the effectiveness of the constructed STOA-VMD algorithm, which shows its high efficiency in signal feature extraction and damage pattern recognition, and further realizes the damage state classification of polymer composites.

Comment 3:Equation 2, the meaning of the operator < > should be explicitly stated.

Response: Thanks for your valuable suggestion.We feel sorry to trouble you in reading.We have explained some operators, please see page 2, lines 79-86.

Comment 4:Optimising the VMD settings (K and alfa) is still a very relevant issue. The Authors propose here the use of the Sooty Tern Optimization Algorithm (STOA) for such purposes. However, this return two (slightly) different settings for sample 1 and 2. This Reviewer's issue is what would happen for a larger number of samples, i.e. if the averages would remain mostly similar or diverge.

Just as a note, the final result, alfa = 2000, is in line with what one would generally expect, i.e. something around alfa = 2950 for moderate bandwidth constraining.

Response:Thanks for the reviewer’s comments.The influence of optimizing K and alfa in VMD settings on a large number of samples mainly depends on the signal characteristics of different samples and the decomposition methods used. In general, for a large number of signal samples, adjusting the K and alfa in the VMD setting may cause the average value to change, or the difference between the samples may increase or decrease.

Specifically, when we increase the K value, the number of modes decomposed by VMD will increase, so the time locality and frequency locality characteristics of the signal can be better preserved. This may make the average value of some samples change greatly, especially for those signals with strong time and frequency changes.

On the other hand, when we increase the alfa value, the bandwidth limitation of VMD becomes narrower, resulting in a narrower range of decomposed modal bands. This helps to improve the accuracy of signal decomposition, especially when dealing with signals with noise components such as Gaussian noise. However, this may also reduce the average of some signals, especially those with relatively wide bandwidth.

In general, selecting the appropriate VMD setting value requires comprehensive consideration of signal characteristics and specific processing applications. When processing a large number of signal samples, it is necessary to optimize according to the situation and analyze to confirm whether the final result is in line with expectations. Therefore, in view of the diversity and complexity of the damage signal of composite materials, this paper selects the average value according to the comparison sample 1 and sample 2, and applies it to the damage signal processing of polymer composite materials in the manuscript.

Comment 5:The paper should provide more details about the AE monitoring system and, in general, more context for Acoustic Emission-based Structural Health Monitoring, with applications to fiber-reinforced polymers (such as in c) but also other materials.

Response:Thanks for the reviewer’s comments.As suggested by the reviewer,We have supplemented the relevant literature and some contents in the reviseript.d manusc.Please see page 2, lines 51-60.

Minor and editorial issues:

Comment 1:Figure 3 seems to be ‘stretched’, plus the font size changes for different writings and some of them are too small to be easily readable.

Response: We sincerely thank the reviewer for this suggestion and careful reading.We have modified the graphics.Please see page 8, lines 254. We sincerely hope to meet the graphic standards of journals.

Comment 2: In general, for several figures (e.g. Figures 7 and 11) the font size of the x- and y-axis labels and ticks and/or the legend is too small.

Response:Thanks for the reviewer’s comments.We 're sorry again for the bad experience you had with reading.We modified the label, font, length and other issues in the whole manuscript graphics, and further improved the clarity of the graphics.

Comment 3:Table 5: is it really needed to report the percentage values up to the third decimal digit? That means an accuracy of 10 parts per million, which seems to be a bit excessive.

Response:Thank you for the above suggestions.We believe that this accuracy is not a problem. Because this algorithm is based on the STOA optimization method of VMD, it has such high recognition ability of damage state and also has such good experimental results. In addition, there are many other areas of the use of VMD recognition algorithm in the literature is also accurate to the decimal point after four, such as this literature (https://doi.org/10.3390/app10113674 ) using VMD to identify the deformation stage and crack initiation of TC11 alloys, the correct judgment rate of crack is 96.33 % . And another literature ( https://doi.org/10.1155/2023/1939606 ) studies the application effect of signal denoising algorithm based on VMD-Hilbert transform in the leakage position of water supply pipeline and the recognition accuracy error is also accurate to the percentage value of four digits after decimal point.

Comment 4:It is not clear why the parameter K (uppercase) is sometimes written as k (lowercase).

Response:Thanks for the reviewer’s comments.We feel very sorry for our negligence to cause you such trouble. We have changed all the parameter K in the manuscript including the text and the formula to uppercase.

We would like to thank the reviewers again for taking the time to review the manuscript patiently and carefully.

Yours Sincerely,

Luying Zhang.

E-mail:zlyyy2023@163.com

Round 2

Reviewer 1 Report

Authors have addressed all the queries. Article may be accepted in its present form.

Author Response

Dear reviewer,

Thank you very much for your reading our paper carefully and giving the above positive comments.

Comment 1:Authors have addressed all the queries. Article may be accepted in its present form.

Response: Thank you very much for your reading our paper carefully and giving the above positive comments.Thank you for your recognition and understanding.

We would like to thank the reviewers again for taking the time to review the manuscript patiently and carefully.

Yours Sincerely,

Luying Zhang.

E-mail:zlyyy2023@163.com

Reviewer 3 Report

After having carefully read this last version of the manuscript, it can be said that the original paper has been correctly amended and improved by the Authors, providing adequate answers and addressing almost all the emerging issues.

The manuscript is interesting and fits well with the aim of the "Polymers" Journal. However, there are still some mistakes that need to be corrected in the manuscript.

The most important among all concerns the writing of the Authors names in the references or in the text, and some references that do not coincide with what is written in the manuscript.

For example:
- Page 2, line 50, "Then Leandro F et al. [18] applied acoustic emission technology ...", does not coincide with the reference [18];
- Page 2, line 55, "... Djabal et al. [19] ...", must be Djabali et al.;
- Page 2, line 80,"... Civera M et al. [23] ...", does not coincide with the reference [23];
- etc, etc ...

Please correct all these errors and others, which surely exist in the text, before the article is published.

According to what was said above, the Reviewer's opinion is that the manuscript can be accepted for publication after the described minor revisions.

Moderate editing of the English language is required.

Author Response

Dear reviewer,

Thank you very much for your reading our paper carefully and giving the above positive comments. We are sorry that our carelessness has brought you a bad reading experience. Based on your suggestion and request, we have made corrected modifications on the revised manuscript. Meanwhile, the manuscript had be reviewed and edited by a language service company. We hope that our work can be improved again. Furthermore, we would like to show the details as follows:

Comment 1:The most important among all concerns the writing of the Authors names in the references or in the text, and some references that do not coincide with what is written in the manuscript.

For example:

- Page 2, line 50, "Then Leandro F et al. [18] applied acoustic emission technology ...", does not coincide with the reference [18];

- Page 2, line 55, "... Djabal et al. [19] ...", must be Djabali et al.;

- Page 2, line 80,"... Civera M et al. [23] ...", does not coincide with the reference [23];

- etc, etc ...

Response: Thanks for your careful checks. We are sorry that our carelessness has brought you a bad reading experience. We have completed these correct writing and referencing problems and have reviewed them again. Please see page 2, lines 52 and 80; page 5, lines 183; page 16, lines 449 and 459; page 16, lines 463; page 17, lines 489.

We would like to thank the reviewers again for taking the time to review the manuscript patiently and carefully.

Yours Sincerely,

Luying Zhang.

E-mail:zlyyy2023@163.com
